# POSE-GUIDED MOTION DIFFUSION MODEL FOR TEXT-TO-MOTION GENERATION

## ABSTRACT

3D Human motion generation, especially textual conditioning motion generation, is a vital part of computer animation. However, during training, multiple actions are often coupled within a single textual description, which complicates the model's learning of individual actions. Additionally, the motion corresponding to a given text can be diverse, which makes it difficult for the model learning and for the user to control the generation of motions that contain a specific pose. Finally, motions with the same semantics can have various ways of expression in the forms of texts, which further increases the difficulty of the model's learning process. To solve the above challenges, we propose the Pose-Guided Text to Motion (PG-T2M) with the following designs. Firstly, we propose to divide the sentences into sub-sentences containing one single verb and make the model learn the specific mapping from one single action description to its motion. Secondly, we propose using pose priors from static 2D natural images for each sub-sentence as control signals, allowing the model to generate more accurate and controllable 3D pose sequences that align with the sub-action descriptions. Finally, to enable the model to distinguish which sub-sentences describe similar semantics, we construct a pose memory storing semantic-similar sub-sentences and the corresponding pose representations in groups. These designs together enable our model to retrieve the pose information for every single action described in the text and use them to guide motion generation. Our method achieves state-of-the-art performance on the HumanML3D and KIT datasets.

## 1 INTRODUCTION

Text-to-motion generation aims to generate human motion sequences given a textual description, which has a wide range of applications in game design, animation, robotics, and other fields. Recently, there has been rapid development in text-to-motion generation, where the recent methods typically use text directly as a condition, employing diffusion models to control motion generation. However, these methods overlook the following three problems.

*Firstly, as shown in Figure 1a, the training data usually contains complicated texts including multiple actions, wherein these coupled actions make it challenging for the model to learn the mapping between individual actions and their corresponding textual segments.* For example, multi-action data accounts for more than 60% in the HumanML3D (Guo et al., 2022) dataset, empirically we observe that the existing models like MDM (Tevet et al., 2023) and MotionDiffuse (Zhang et al., 2022) trained with these data usually coupled with multiple actions has a significant performance drop on a single action in Figure 1b. *Secondly, as shown in Figure 1c, motions described by the same text (such as 'kicks') can be highly diverse. This increases the uncertainty of motion generation, making it difficult for the user to control the generation of motions that contain a specific pose.* Motion is a sequence of poses and pose information plays a vital role in motion generation. Therefore, we believe that introducing prior knowledge of the diverse poses corresponding to a single action description can provide more prior information related to the textual description, making the results better aligned with the condition. Since the pose-prior is easier to obtain through off-the-shelf text-to-image and image-to-pose models, we choose to use several static poses rather than a continuous motion sequence as the guide. *Finally, as illustrated in Figure 1d, due to the diversity in natural language, motions with the same semantics can have various ways of expression in the forms of texts.* This diversity increases the difficulty of the model's learning process. If we can assist the model in

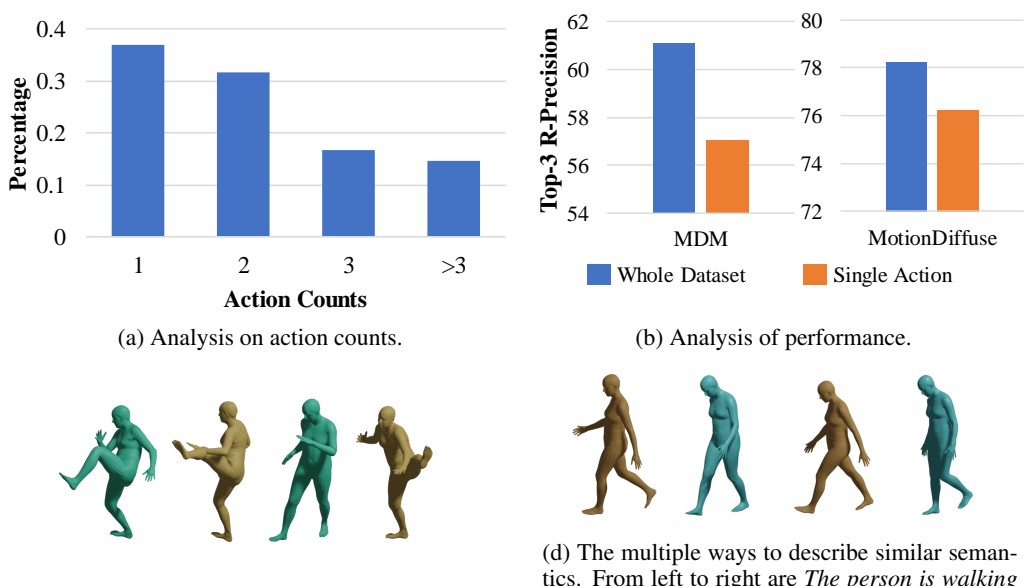

(a) Analysis on action counts.

(b) Analysis of performance.

(c) The diverse motions we corresponding to *a person kicks*. The same verb *kick* refers to different motions.

(d) The multiple ways to describe similar semantics. From left to right are *The person is walking normally forward; The person is doing a casual walk; The person is walking at a normal pace; A person takes several steps*.

Figure 1: (a)Analysis of data distribution of HumanML3D dataset about texts including multiple actions. (b)Performance comparison on single-action texts on HumanML3D. (c)The diversity of motions described by the same text description. (d) Some synonymous texts may describe similar semantics.

establishing clusters of the action descriptions, that is, informing the model which text semantics are closer, it could better aid the model's learning process.

To address the above issues, we propose a Pose-Guided Text to Motion (PG-T2M) model, which constructs a pose memory that stores mappings between text clusters of semantically similar single action and their pose information. This enables the model to acquire prior information related to the pose of all actions within the text when generating motions. Firstly, to facilitate the model in better learning the correspondence between individual action descriptions in complex text input and motion, we utilize a sentence parser to break down the complex text into several sub-sentences, each containing only one action. Then, to acquire the pose information of a single action, we use the text-to-image diffusion model to acquire the image corresponding to the sub-sentence and use pose extractors to obtain pose features from the image. Finally, to further aid the model in learning the semantic-similar texts, we cluster all the sub-sentences in the training set and maintain a pose memory that stores pose features corresponding to each cluster. As sub-sentences within the same cluster possess similar semantics, we retain pose features only for a few sub-sentences within each cluster for efficiency. During training and inference, given a text, we also parse it into sub-sentences and retrieve the pose features from the pose memory. As we use the static pose feature as prior information, we also design a temporal encoder to further encode the temporal relationship of those pose features. These encoded pose features are then utilized alongside text features to control the generation of motion.

In summary, our contributions are:

- We propose a Pose-Guided Text to Motion (PG-T2M) model, which is the first to introduce pose information in the text-to-motion generation. We introduce the pose information from a large-scale text-to-image diffusion model to provide various poses prior related to the text, thus helping the model generate motions better aligned with the text conditions.

- We propose to parse the complex text prompt into sub-action descriptions to help the model better learn the correspondence between each sub-action description and the motion. We leverage text-to-image diffusion models and pose extractors to automatically obtain pose

representations related to the sub-actions to control motion generation. We also construct a text-pose pose memory to store different texts describing similar semantics and different poses from the same texts to help the model learn the mapping between texts and the poses.

- We achieve state-of-the-art performance and validate the effectiveness of the method on KIT (Plappert et al., 2016) and HumanML3D (Guo et al., 2022) datasets.

## 2 RELATED WORK

**Text to Motion Generation.** Recently, widespread attention has been paid to 3D human motion generation. Some works (Lee et al., 2023; Athanasiou et al., 2022) have attempted to control the generation of human motions using atomic action labels as conditions, but these methods struggle to satisfy the need for generating diverse and complex motion sequences. Thus, an increasing number of researchers are exploring the use of free-form text as a condition to control motion generation. For example, TEMOS (Petrovich et al., 2022), T2M-GPT (Zhang et al., 2023b) and MoMask (Guo et al., 2023) propose encoder-decoder or VAE-based pipelines to generate motions. Recently, diffusion model (Ho et al., 2020) have been introduced to text-to-motion generation by MDM (Tevet et al., 2023), MotionDiffuse (Zhang et al., 2022), MLD (Chen et al., 2023), GraphMotion (Jin et al., 2023), etc., due to its outstanding ability in many generative tasks. These methods often directly encode the features of complex text prompts to control the generation of actions, resulting in the model's difficulty in learning the correspondence between individual action descriptions and motion from complex text prompts. They also overlook the motions described by the same text can be highly diverse, which increases the uncertainty of motion generation, making it difficult for the user to control the generation of motions that contain a specific pose. Thus, we propose to divide the sentence into sub-sentences containing only one single verb and introduce the pose information of each verb as the additional control signal by constructing a pose memory that stores mappings between text clusters of semantically similar single action and their pose information. This enables the model to acquire prior information related to the pose of all sub-actions within the text for generation. One related work to ours is MAA (Azadi et al., 2023), which introduces pose information by simply pre-training the model on text-to-pose datasets and fine-tuning it on text-to-motion data. However, it failed to establish a connection between the pose caption and the motion description, resulting in the model still struggling to understand the correspondence between each sub-action in the motion description and the pose. In our work, we directly utilize the descriptions of each sub-action to retrieve the most relevant poses from the pose memory, explicitly using these poses as control signals, allowing the model to generate more precise and controllable action sequences.

**Codebook and Memory Bank.** Codebook, memory bank, or other memory-based approaches have been widely used in image classification (He et al., 2020), multimodal alignment (Duan et al., 2022), and generative models (Van Den Oord et al., 2017). For example, (Cao et al., 2017) used codebook to speed up image retrieval. However, such a mechanism is yet to be fully explored in the field of motion generation. We are the first to apply the concept of a memory bank to the text-to-motion generation, aiming to establish a mapping from text descriptions to corresponding pose information and enable the model to retrieve pose information to control the generation of motion. One related work to ours is RemoDiffuse (Zhang et al., 2023c), which retrieves real motion using text from the entire training set to control the motion generation. However, their retrieval source is limited to the training set and requires the same distribution between training and test data. Moreover, retrieving the whole motion sequence also neglects fine-grained sub-actions in motion descriptions. In contrast, our pose information in the memory comes from the open-world image generation model, which allows the pose memory to obtain knowledge beyond the training set. In addition, our memory stores the correspondence between the poses of sub-actions and their descriptions, rather than the complete motion sequences, allowing the model to learn more fine-grained pose priors for each sub-actions.

## 3 METHOD

### 3.1 OVERVIEW

Given a text input, our goal is to generate a motion sequence $x^{1:N}$ that corresponds to the text input, where $N$ represents the length of the motion.

The overview of our method is presented in Figure 2. Our pipeline contains three modules: the **Pose Memory Construction** module to construct a pose memory storing the pose features of semantic-similar single actions, the **Pose Guided Conditioning** to retrieve the pose features from the pose memory and encode the temporal relations between them as the control signal, and the **Motion Diffusion** module to generate the motions based on the conditions using the diffusion model.

Specifically, we adopt **Motion Diffusion** model MDM (Tevet et al., 2023) as our baseline, which encodes the text prompt as a condition and uses the diffusion model to generate motions based on the conditions. More details of the baseline are described in Section 3.2. The **Pose Memory Construction** described in Section 3.3 constructs a pose memory that stores mappings between text clusters of semantic similar single actions and their pose information. Firstly, to enable the model to learn the relationship between a single action description and its motion from complex texts, we propose to split the texts into sub-sentences that only contain one action. Then, to acquire the pose information of a single action for a more controllable motion generation, we propose a pipeline to generate pose representations from sub-sentences using off-the-shelf text-to-image generative models and pose detectors. Since different sub-sentences may describe similar semantics, we cluster the sub-sentences in the training set to aid the model in learning the semantic-similar sub-sentences. We then construct a pose memory that stores the mapping from the clusters to the pose features. The **Pose Guided Conditioning** described in Section 3.4 uses the pose information to help the model with motion generation. Firstly, we retrieve the pose features corresponding to the sub-sentences from the pose memory. Since there exists a sequential relationship between pose features corresponding to different sub-sequences, we use a temporal encoder to further encode these pose features. The encoded features are then utilized alongside text features to control the generation.

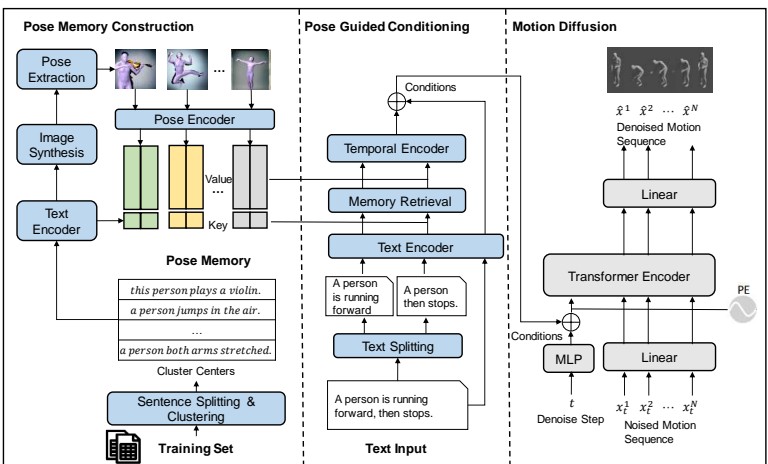

Figure 2: **The overall framework. Left: Pose Memory Construction.** We split text in the training set into sub-sentences and cluster them as the keys of the pose memory. Then we synthesize the images from the sub-sentences and extract the pose from the images, which serves as the values of the pose memory. **Middle: Pose Guided Conditioning.** The model first encodes the split sub-sentences and uses them to retrieve the corresponding pose feature from the pose memory. To further encode the temporal relationship between pose features, a temporal encoder is involved. The encoded pose feature and original text feature are added together as guidance. **Right: Motion Diffusion.** During inferring, the model is provided with conditions and starts from pure noise to predict the motions using the diffusion model following MDM. In each step, the model predicts the final motion sequence $\hat{x}_0^{1:N}$ from input motion sequence $x_t^{1:N}$ guided by the condition.

## 3.2 MOTION DIFFUSION MODEL (MDM) REVISIT

Diffusion models are a new type of generative model that has an outstanding ability to tackle image-generation tasks. Recently, it has been proved by several works (Zhang et al., 2022; Tevet et al., 2023) that diffusion models also perform well in motion generation. In this paper, we use MDM (Tevet et al., 2023), a diffusion-based model, as our baseline model.

MDM follows a Markov chain to add Gaussian noise to the original motion sequence $x^{1:N}$ to make it pure noise in:

$$q(x_t|x_{t-1}) = \mathcal{N}(\sqrt{\alpha_t}x_{t-1}, (1-\alpha_t)I), \tag{1}$$

where $\alpha_t$ is hyper-parameters, and we use $x_t$ to represent $x_t^{1:N}$ for simplicity, which is the motion sequence after adding noise for $t$ times. Then, we can efficiently obtain $x_t$ from $x_0$ following (Ho et al., 2020) by:

$$q(x_t|x_0) = \sqrt{\overline{\alpha}_t}x_0 + \sqrt{1-\overline{\alpha}_t}\epsilon, \tag{2}$$

where $\overline{\alpha}_t = \prod_{m=1}^{t} \alpha_m$ and $\epsilon \sim \mathcal{N}(0, I)$.

For the reverse process, timestep $t$ is fed into an FFN and added to text features $c$ to obtain the guidance token $z$. MDM dose not predict $x_{t-1}$ or noise $\epsilon_t$ from $x_t$. Instead, it directly predicts the final result $x_0$ use a transformer encoder $F$: $\hat{x}_0 = F(x_t, z)$, where $z$ is the guidance token. They use MSE loss to optimize the diffusion model:

$$\mathcal{L} = \mathbb{E}_{x_0 \sim q(x_0|c), t \sim [1,T]}[\|x_0 - \hat{x}_0\|_2^2] \tag{3}$$

In the training process, timestep $t$ is randomly chosen and this reverse process is performed once. During testing, after obtaining $\hat{x}_0$ from $x_T$, MDM diffuses it back to $x_{T-1}$, and this process will be iterated for $T$ times and finally obtain $\hat{x}_0$ from $x_1$, which will be the ultimate output.

### 3.3 POSE MEMORY CONSTRUCTION

To provide the model with pose priors for each sub-action in the text, enabling more precise and controllable action generation, we propose to split texts into sub-sentences only containing a single verb and obtain pose-related priors for each sub-sentence. Due to various descriptions for the same semantic meaning using natural language, to enhance the model's learning of semantically similar sub-sentences, we propose to employ a pose memory to store the mapping from semantic-similar sub-sentences to their pose features.

**Text Splitting.** As previous models do not pay enough attention to the inner structure of the input texts, especially they may include multiple actions that make text-conditional motion generation more difficult, our proposed text splitting tries to tackle the problem. Specifically, we split the sentence into shorter texts with only one action to disentangle the multiple verbs in the texts. We consider the predicate-argument structure of the input sentence to split it into sub-actions. We first use off-the-shelf SRLBert (Shi & Lin, 2019) to obtain the PropBank-style (Palmer et al., 2005) semantic role label of the sentences. We split the sentence into sub-actions based on the verb tag [V]. Each verb has some attached tags, for example, [ARG0] represents *Proto-Agent* of the verb and [ARG1] stands for *Proto-Patient* of the verb. We keep each verb and the attached tags as the split sub-actions. After splitting, we use a text encoder $\text{T}(\cdot)$ to compute the text features of the sentence and sub-actions.

**Pose Feature Generation.** As it is hard to obtain a massive text-pose paired dataset from scratch, we choose to probe knowledge from trained generative models. Stable Diffusion (Rombach et al., 2021), which has proved its ability to generate diverse images, is leveraged by us to generate pose-related images from our text-only data. Specifically, after obtaining the split sub-sentences in the training set, we use the Stable Diffusion to generate $m_p$ images from each sub-sentence. The images generated are then fed into PyMAF-X (Zhang et al., 2023a), a pre-trained pose extractor, to extract the SMPL (Loper et al., 2023) pose annotation from these images. This allows us to obtain $m_p$ pose annotations for each sub-action. The reason for generating multiple poses is that the same sub-action can encompass various postures, and we aim to retain this diversity. Note that we use the static pose feature from synthesized images instead of the dynamic motion feature from synthesized videos because the text-to-image model typically exhibits higher generalization due to massive training data and is more efficient than the text-to-video model. To consider the temporal relationship of the static pose features, we also design a temporal encoder which will be discussed later.

To speed up the training and inferring process, the pose features from texts are generated in an offline way instead of along with the model, and stored for later use.

**Pose Memory Construction.**

To obtain the semantic-similar sub-sentences, we use k-means to cluster the split sub-sentences from the training set into $k$ clusters by their text features. Then, to maintain variety within the text of each

cluster, $m_t$ texts are randomly selected and fed into the text-image-pose pipeline for each cluster. Thus, the total number of texts is $N_t = m_t \times k$. To enhance variety, $m_p$ images will be generated from each selected text and the number of poses is $N_p = m_p \times N_t$. The text features in each cluster along with their cluster center serve as the key, and the corresponding poses are the values. So our key in the pose memory is

$$\text{Key} = \text{T}(s_1, s_2, ..., s_{N_t}) \in \mathbb{R}^{N_t \times d_t}, \tag{4}$$

where $s_1, s_2, ..., s_{N_t}$ are the sub-sentences in the pose memory, $\text{T}(\cdot)$ is the text encoder, and $d_t$ is the dimension of the text feature space. And our value in the pose memory is the corresponding pose representations $\text{Value} \in \mathbb{R}^{N_p \times d_p}$, where $d_p$ stands for the pose dimensions. Note, our pose memory provides a one-to-multiple mapping for the diversity of pose features, where a single text can be mapped to $m_p \times m_t$ poses, and the selection of pose features will be discussed in Section 3.4.

### 3.4 Pose Guided Conditioning.

After obtaining the pose memory, we can retrieve the pose feature of the split sub-sentences during training and inference to guide the motion generation. Since there exists a temporal relationship between poses (such as the temporal order in which each pose appears), we propose a temporal encoder to further encode these pose features.

**Pose Memory Retrieval.** Given a sub-sentence, the text feature of it $f_{query} \in \mathbb{R}^{d_t}$ is used to compute the distance between itself and text features $f_{key} \in \mathbb{R}^{d_t}$ stored in our pose memory. Then, the fed-in sub-sentence is classified in one certain cluster. Later, we randomly choose a pose in this cluster as the corresponding pose representation. Noting that here we select the pose value from the whole cluster ($m_t \times m_p$ poses) for variety, instead of selecting from only those corresponding with that certain key ($m_p$ poses). This design helps enhance the variety of the motion generated.

**Temporal Encoder.** The pose retrieved from the pose memory is extracted from static images and there exists a temporal relationship such as sequential relationship between pose features corresponding to different sub-sequences. Therefore, we use a transformer encoder as the temporal encoder to further encode these pose features. We set the max number of sub-sentences to $N_s$. Suppose that the original sentence is split into $n_s$ sub-actions, if $n_s < N_s$, we will repeat the fetched poses in an interleaving way. For instance, if $n_s = 2, N_s = 4$, we repeat action $AB$ into $AABB$. Otherwise, we simply truncate it. These $N_s$ poses are fed into the temporal encoder. Suppose the pose representations are $\boldsymbol{P} \in \mathbb{R}^{N_s \times d_p}$, we process it as:

$$f_p = \text{E}(\boldsymbol{P} + \text{PE}(\boldsymbol{P})), \tag{5}$$

where $\text{E}(\cdot)$ is the transformer encoder and $\text{PE}(\cdot)$ is the positional embedding.

To control the generation of motions with both text and pose features, we add $f_p$ to the original text feature $f_t$ of the sentence to get the final hybrid feature:

$$f_{hybrid} = f_p \oplus f_t, \tag{6}$$

where $\oplus$ stands for vector addition. The hybrid feature $f_{hybrid}$ is used to replace the text feature $c$ in the diffusion part described in Section 3.2 as the control signal.

Our training and inferring processes are similar to what is described in Section 3.2, and our objective function is the same as Equation (3). During inferring, the diffusion process will be iterated for $T$ times. The randomness in pose fetching is kept, but the pose fetched in the first iterations will not be changed between iterations and we do not repeat the retrieval in the later iterations.

## 4 Experiments

### 4.1 Datasets and Evaluation Metrics

**Datasets**. In this study, we employ the HumanML3D dataset (Guo et al., 2022) and the KIT dataset (Plappert et al., 2016) to assess the effectiveness of the proposed approaches in the task of text-to-motion generation, following (Guo et al., 2022; Tevet et al., 2023; Zhang et al., 2023c). HumanML3D (Guo et al., 2022) is a widely-used dataset in the text-to-motion domain recently, which provides 14616 motions with 44970 text annotations. KIT (Plappert et al., 2016) is another widely used dataset including 3911 motions annotated by 6353 textual descriptions.

**Evaluation Metrics**. We employ the following four metrics adopted by (Guo et al., 2022; Zhang et al., 2022; Tevet et al., 2023). *1) R-precision (R)*. For each textual prompt and its corresponding generated motion, 31 other pairs are randomly selected. The matching of them is computed and the top-k accuracy is obtained. *2) Multi-Modal Distance (MM Dist)*. Using the pre-trained contrastive model, we compute the distance between the input text and the generated motion. *3) FID*. We use FID on the features extracted from ground truth and generated motions to measure the distribution distance. *4) Diversity*. We randomly separate the motions generated into pairs and compute the joint differences of each pair to show the variety of generated motions.

Table 1: **Quantitative evaluation results on HumanML3D (Guo et al., 2022) dataset.** $\pm$ indicates 95% confidence interval. An up-arrow↑ indicates the performance is better if the value is higher. We use **bold** to represent the best result in the table. †: ReMoDiffuse has access to the training set samples during evaluation. MoMask does not apply diversity as a metric.

| Method | R% (top 1)↑ | R% (top 2)↑ | R% (top 3)↑ | FID↓ | MM Dist↓ | Diversity - |
|---|---|---|---|---|---|---|
| Ground Truth | $51.1^{\pm.3}$ | $70.3^{\pm.3}$ | $79.7^{\pm.2}$ | $0.002^{\pm.002}$ | $2.974^{\pm.008}$ | $9.503^{\pm.065}$ |
| TEMOS (Petrovich et al., 2022) | $42.4^{\pm.2}$ | $61.2^{\pm.2}$ | $72.2^{\pm.2}$ | $3.734^{\pm.028}$ | $3.703^{\pm.008}$ | $8.973^{\pm.071}$ |
| T2M-GPT (Zhang et al., 2023b) | $49.2^{\pm.3}$ | $67.9^{\pm.2}$ | $77.5^{\pm.2}$ | $0.141^{\pm.005}$ | $3.121^{\pm.009}$ | $9.722^{\pm.082}$ |
| MLD (Chen et al., 2023) | $48.1^{\pm.3}$ | $67.3^{\pm.3}$ | $77.2^{\pm.2}$ | $0.473^{\pm.013}$ | $3.196^{\pm.010}$ | $9.724^{\pm.082}$ |
| †ReMoDiffuse (Zhang et al., 2023c) | $51.0^{\pm.5}$ | $69.8^{\pm.6}$ | $79.5^{\pm.4}$ | $0.103^{\pm.004}$ | $2.974^{\pm.016}$ | $9.018^{\pm.075}$ |
| Fg-T2M (Wang et al., 2023) | $49.2^{\pm.2}$ | $68.3^{\pm.3}$ | $78.3^{\pm.2}$ | $0.243^{\pm.019}$ | $3.109^{\pm.007}$ | $9.278^{\pm.072}$ |
| GraphMotion (Jin et al., 2023) | $50.4^{\pm.3}$ | $69.9^{\pm.2}$ | $78.5^{\pm.2}$ | $0.116^{\pm.007}$ | $3.070^{\pm.008}$ | $9.692^{\pm.067}$ |
| MAA (Azadi et al., 2023) | - | - | $67.6^{\pm.2}$ | $0.774^{\pm.007}$ | - | $8.23^{\pm.064}$ |
| BAD(OAAS) (Hosseyni et al., 2024) | $51.7^{\pm.2}$ | $71.3^{\pm.3}$ | $80.8^{\pm.3}$ | $0.065^{\pm.003}$ | $\mathbf{2.901^{\pm.008}}$ | $9.694^{\pm.068}$ |
| BAD(CBS) (Hosseyni et al., 2024) | $51.1^{\pm.2}$ | $70.4^{\pm.2}$ | $80.0^{\pm.2}$ | $0.049^{\pm.003}$ | $2.957^{\pm.006}$ | $9.688^{\pm.089}$ |
| BAMM (Pinyoanuntapong et al., 2024) | $52.2^{\pm.3}$ | $71.5^{\pm.3}$ | $80.8^{\pm.3}$ | $0.055^{\pm.002}$ | $2.936^{\pm.077}$ | $9.636^{\pm.009}$ |
| MDM (Tevet et al., 2023) | $32.0^{\pm.5}$ | $49.8^{\pm.4}$ | $61.1^{\pm.7}$ | $0.544^{\pm.044}$ | $5.566^{\pm.027}$ | $9.559^{\pm.086}$ |
| Ours with MDM | $33.8^{\pm.4}$ | $53.8^{\pm.7}$ | $64.5^{\pm.7}$ | $0.689^{\pm.042}$ | $5.355^{\pm.028}$ | $9.678^{\pm.096}$ |
| MotionDiffuse (Zhang et al., 2022) | $49.1^{\pm.1}$ | $68.1^{\pm.1}$ | $78.2^{\pm.1}$ | $0.630^{\pm.001}$ | $3.113^{\pm.001}$ | $9.410^{\pm.049}$ |
| Ours with MotionDiffuse | $51.0^{\pm.5}$ | $70.0^{\pm.3}$ | $79.6^{\pm.4}$ | $0.151^{\pm.008}$ | $2.977^{\pm.007}$ | $9.401^{\pm.155}$ |
| MoMask (Guo et al., 2023) | $52.1^{\pm.2}$ | $71.3^{\pm.2}$ | $80.7^{\pm.2}$ | $\mathbf{0.045^{\pm.002}}$ | $2.958^{\pm.008}$ | — |
| Ours with MoMask | $\mathbf{53.1^{\pm.4}}$ | $\mathbf{72.0^{\pm.3}}$ | $\mathbf{81.5^{\pm.6}}$ | $0.064^{\pm.009}$ | $2.908^{\pm.017}$ | — |

Table 2: **Quantitative evaluation results on KIT (Plappert et al., 2016) dataset.** $\pm$ indicates 95% confidence interval. An up-arrow↑ indicates the performance is better if the value is higher. We use **bold** to represent the best result in the table. †: ReMoDiffuse has access to the training set samples during evaluation. MoMask does not apply diversity as a metric.

| Method | R% (top 1)↑ | R% (top 2)↑ | R% (top 3)↑ | FID↓ | MM Dist↓ | Diversity - |
|---|---|---|---|---|---|---|
| Ground Truth | $42.4^{\pm.5}$ | $64.9^{\pm.6}$ | $77.9^{\pm.6}$ | $0.031^{\pm.004}$ | $2.788^{\pm.012}$ | $11.08^{\pm.097}$ |
| TEMOS (Petrovich et al., 2022) | $35.3^{\pm.6}$ | $56.1^{\pm.7}$ | $68.7^{\pm.5}$ | $3.717^{\pm.051}$ | $3.417^{\pm.019}$ | $10.84^{\pm.100}$ |
| T2M-GPT (Zhang et al., 2023b) | $41.6^{\pm.6}$ | $62.7^{\pm.6}$ | $74.5^{\pm.6}$ | $0.514^{\pm.029}$ | $3.007^{\pm.023}$ | $10.92^{\pm.108}$ |
| MLD (Chen et al., 2023) | $39.0^{\pm.8}$ | $60.9^{\pm.8}$ | $73.4^{\pm.7}$ | $0.404^{\pm.027}$ | $3.204^{\pm.027}$ | $10.80^{\pm.117}$ |
| †ReMoDiffuse (Zhang et al., 2023c) | $42.7^{\pm1.4}$ | $64.1^{\pm.4}$ | $76.5^{\pm5.5}$ | $0.155^{\pm.006}$ | $2.814^{\pm.012}$ | $10.80^{\pm.105}$ |
| Fg-T2M (Wang et al., 2023) | $41.8^{\pm.5}$ | $62.6^{\pm.4}$ | $74.5^{\pm.4}$ | $0.571^{\pm.047}$ | $3.114^{\pm.015}$ | $10.93^{\pm.083}$ |
| GraphMotion (Jin et al., 2023) | $42.9^{\pm.7}$ | $64.8^{\pm.6}$ | $76.9^{\pm.6}$ | $0.313^{\pm.013}$ | $3.076^{\pm.022}$ | $11.12^{\pm.135}$ |
| BAD(OAAS) (Hosseyni et al., 2024) | $41.7^{\pm.6}$ | $63.1^{\pm.6}$ | $75.0^{\pm.6}$ | $0.221^{\pm.012}$ | $2.941^{\pm.025}$ | $11.000^{\pm.100}$ |
| BAD(CBS) (Hosseyni et al., 2024) | $40.8^{\pm.4}$ | $61.2^{\pm.7}$ | $73.4^{\pm.7}$ | $0.246^{\pm.019}$ | $3.100^{\pm.021}$ | $10.874^{\pm.083}$ |
| BAMM (Pinyoanuntapong et al., 2024) | $43.6^{\pm.7}$ | $66.0^{\pm.6}$ | $79.1^{\pm.5}$ | $0.200^{\pm.011}$ | $2.714^{\pm.016}$ | $10.914^{\pm.097}$ |
| MDM (Tevet et al., 2023) | $16.4^{\pm.4}$ | $29.1^{\pm.4}$ | $39.6^{\pm.4}$ | $0.497^{\pm.021}$ | $9.19^{\pm.022}$ | $10.847^{\pm.109}$ |
| Ours with MDM | $19.5^{\pm.4}$ | $33.9^{\pm.5}$ | $43.5^{\pm.5}$ | $0.365^{\pm.041}$ | $9.042^{\pm.015}$ | $10.808^{\pm.093}$ |
| MotionDiffuse (Zhang et al., 2022) | $41.7^{\pm.4}$ | $62.1^{\pm.4}$ | $73.9^{\pm.4}$ | $1.954^{\pm.062}$ | $2.958^{\pm.005}$ | $\mathbf{11.10^{\pm.143}}$ |
| Ours with MotionDiffuse | $42.5^{\pm.6}$ | $64.9^{\pm.8}$ | $77.7^{\pm.9}$ | $\mathbf{0.113^{\pm.021}}$ | $2.797^{\pm.009}$ | $10.91^{\pm.069}$ |
| MoMask (Guo et al., 2023) | $43.3^{\pm.7}$ | $65.6^{\pm.5}$ | $78.1^{\pm.5}$ | $0.204^{\pm.011}$ | $2.779^{\pm.022}$ | — |
| Ours with MoMask | $\mathbf{44.6^{\pm.6}}$ | $\mathbf{67.1^{\pm.7}}$ | $\mathbf{79.8^{\pm.6}}$ | $0.143^{\pm.006}$ | $\mathbf{2.608^{\pm.009}}$ | — |

## 4.2 IMPLEMENTATION DETAILS

For the pose feature generation, we use the Stable Diffusion 2.1 (Rombach et al., 2021) to generate images from text and use PyMAF-X (Zhang et al., 2023a) to extract pose from images. For the motion diffusion model, we use *CLIP-ViT-B/32* (Radford et al., 2021) as the text encoder. We train the model for both HumanML3D and KIT with an 8-layer transformer in motion encoder, a 2-layer transformer in pose temporal encoder, and a batch size of 64. For the pose memory, we set $k = 2048, m_t = 16, m_p = 12$ for HumanML3D and $k = 512, m_t = 8, m_p = 6$ for KIT. For our MotionDiffuse (Zhang et al., 2022)-based implementation, we set $k = 1024$ for HumanML3D and $k = 256$ for KIT. The latent dimensions of the hybrid feature and motion encoder are 256 and 512. For the noising process, we set $T = 1000$ with a cosine noise schedule in the training stage. We train three models based on MDM (Tevet et al., 2023), MotionDiffuse (Zhang et al., 2022) and MoMask (Guo et al., 2023), respectively. More details are provided in Appendix A.3.

## 4.3 COMPARISON WITH STATE-OF-THE-ARTS

**Comparisons on HumanML3D and KIT.** The results on HumanML3D and KIT are shown in Table 1 and Table 2. Furthermore, we gave qualitative comparisons in Appendix A.2.1.

(1) We can see that our model achieves state-of-the-art results in the two datasets on R-precisions, which proves that our design increases the correctness of conditional generation.

(2) We plug our module into MDM (Tevet et al., 2023), MotionDiffuse (Zhang et al., 2022), and MoMask (Guo et al., 2023), respectively, and both work much better than the original design. This proves that our approach is generic.

(3) The FID suffers from a slight drop on HumanML3D with some baselines. As we employ a pose memory and poses generated by StableDiffusion (Rombach et al., 2021), it is reasonable that the distribution of our generated samples slightly differs from the distribution of the dataset.

(4) The diversity of our method is slightly lower than the baseline, primarily because we introduced pose information as an additional control condition. As shown in Fig. 3, the model generates motions that better align with the pose conditions, which reduces the diversity. However, this enhances the quality and controllability of the motion generation, as reflected in the improvement of the R-precision. Furthermore, Wang et al. (2023) has also pointed out that higher diversity is not always better, as random motions have greater diversity but very low quality. We also find that the diversity of our generated motions is relatively close to the diversity of ground truth as shown in Table 1, which is reasonable.

**Analysis on Multi-action Entanglement.**

We design an additional experiment to further investigate whether our model performs better in handling the multi-action entanglement in complex texts. From the training set of HumanML3D (Guo et al., 2022), we select those with multiple actions and only use them as training data. For evaluations, we use those with only one single action in the test set. We compare the performance of three baselines (Tevet et al., 2023; Zhang et al., 2022; Guo et al., 2023) w/ or w/o our design in Table 3. The results prove that our model learns each specific action in a highly entangled training set and generates it correctly. This also indicates that our approach, which involves breaking down complex text into sub-sentences and obtaining corresponding pose features through a text-to-motion diffusion model, does not rely on the distribution of training data and exhibits better generalization.

## 4.4 ABLATION STUDIES AND DISCUSSIONS

To further explore if our proposed pipeline is effective and how the hyper-parameters in it will affect the performance, we design several ablation studies. All of these studies are based on the HumanML3D (Guo et al., 2022) dataset using the MDM-based model. Despite the experiment done with $k = 2048$, all ablations are conducted with $k = 1024$ for efficiency. We demonstrate the quantitative results in Tables 4 and 5, from which we can conclude the influence of:

**Design Choices of Pose Memories Construction.** In Table 4, we investigate how different ways of constructing the pose memory influence the results.

Table 3: **Evaluation results on resplit HumanML3D (Guo et al., 2022) dataset.** We choose the multi-action data in the training set to train and use single-action data in the test set to evaluate.

| Method | R% (top 3)↑ | FID↓ | MM Dist↓ |
|---|---|---|---|
| Ground Truth | 74.7 | 0.003 | 3.312 |
| MDM (Tevet et al., 2023) | 53.3 | 0.721 | 5.891 |
| Ours with MDM | **61.8** | **0.717** | **5.517** |
| MotionDiffuse (Zhang et al., 2022) | 60.1 | 0.728 | 3.759 |
| Ours with MotionDiffuse | **72.0** | **0.293** | **3.505** |
| MoMask (Zhang et al., 2022) | 62.5 | **0.150** | 3.631 |
| Ours with MoMask | **73.2** | 0.156 | **3.502** |

Table 4: Ablation study on pose memories construction on HumanML3D (Guo et al., 2022).

| Method | R% (top 3)↑ | FID↓ | MM Dist↓ |
|---|---|---|---|
| Ours($k = 2048$) | **64.5** | **0.689** | **5.355** |
| Ours($k = 1024$) | 64.3 | 0.695 | 5.358 |
| Ours($k = 512$) | 62.6 | 0.818 | 5.404 |
| Ours($k = 256$) | 61.9 | 0.940 | 5.529 |
| Ours($m_t = 16$) | **64.3** | **0.695** | **5.358** |
| Ours($m_t = 8$) | 64.1 | 0.754 | 5.433 |
| Ours($m_t = 4$) | 61.3 | 0.779 | 5.556 |
| Ours($m_p = 12$) | **64.3** | **0.695** | **5.358** |
| Ours($m_p = 6$) | 64.0 | 0.782 | 5.381 |
| Ours($m_p = 3$) | 62.1 | 0.762 | 5.414 |
| Ours(clustered by text) | **64.3** | 0.695 | **5.358** |
| Ours(clustered by pose) | 61.5 | **0.608** | 5.564 |

*Size of the pose memory.* In our pose memory, we cluster the texts into $k$ clusters, keep $m_t$ texts for each cluster center and generate $m_p$ pose features for each text. We change $k$, $m_t$, or $m_p$ to discuss how the scale of the pose memory influences the performance. In detail, the number of clusters may influence the granularity of our pose memory, and the number of texts selected or poses for each text may influence the diversity of the texts in a cluster or the poses for a certain text. For the cluster center $k$, a larger $k$ results in better performance. We suppose the reason is that a smaller $k$ results in clusters containing more text that may not be semantically close enough, so the model is more likely to retrieve mismatched poses. However, from $k = 1024$ to $k = 2048$, the improvement is rather limited. We finally choose $k = 2048$, to balance the performance and the scale of the pose memory. The decrease of both $m_t$ and $m_p$ will degrade R-precision and enlarge Multimodal dist. This proves that the scale of the pose memory matters. The larger the pose memory, the higher the diversity of stored text and poses. During training, the model can also learn more pose priors, thereby improving performance. However, continuing to increase $m_t, m_p$ significantly increases the storage and retrieval costs of the pose memory. For efficiency, we ultimately set $m_t = 16, m_p = 12$.

*Clustering the entries by text or pose.* As shown in the last 2 rows of Table 4, we re-cluster the pose memory by the pose representation instead of by the text features. The comparison shows that our original direct way is better for conditional generation. Re-clustering by pose helps the model generate motions more based on the motion itself rather than text conditioning, resulting in generally more realistic motions that enhance FID but are worse for conditional generation. We suppose that though a pose memory re-clustered by pose truly contains more similar poses in each cluster, it increases the difficulty of text retrieval due to the variety of keys in a cluster. Thus, we cluster the pose memory by texts throughout the experiments unless otherwise specified.

**Design Choices of Pose Memory Usage.** In Table 5, we investigate how several different choices during the usage of the pose memory influence performance. We repeat this ablation with the different baseline, showing the results in Appendix A.1.2.

*Disabling the temporal encoder.* We present the results in row 2 of Table 5. We disable the temporal encoder, directly concatenate the pose features, and use a linear layer to encode the pose features. The degradation shows that the temporal information between pose features is important for motion generation and the temporal encoder can efficiently encode the temporal information.

Table 5: Ablation study on choices of pose memory usage on HumanML3D (Guo et al., 2022).

| Temporal encoder | Random selection | Text splitting | R% (top 3)↑ | FID↓ | MM Dist↓ |
|:---:|:---:|:---:|:---:|:---:|:---:|
| ✓ | ✓ | ✓ | **64.3** | **0.695** | **5.358** |
| ✗ | ✓ | ✓ | 59.5 | 0.704 | 5.767 |
| ✓ | ✗ | ✓ | 60.9 | 0.809 | 5.470 |
| ✓ | ✓ | ✗ | 61.4 | 1.001 | 5.564 |

*Random selection or not.* We randomly choose poses in the cluster during training. During inference, after selecting a pose, it would not be changed during iterations. In the third row of Table 5, we test the performance with randomness disabled during both training and inferring, i.e., choosing a fixed text and pose feature in a cluster. The performance without random selection suffers from a drop. The comparison illustrates that with a random selection, the model is provided with more diverse pose priors during training and inference and works better.

*Disabling the text splitting module.* As shown in the last row in Table 5, we disable the text splitting module and use the raw text to retrieve the pose. Furthermore, we gave visualizations of this ablation study in Appendix A.2.3. This increases FID, Multimodal Dist, and decreases R, showing that using the whole text for retrieval leads to the disability of fetching poses in a fine-grained way, and no static poses could be perfect for multi-action queries. Thus, the obtained pose might be limited and misleading and cannot guide the model to generate motions like real humans.

### 4.5 CONTROL THE MOTION BY POSE

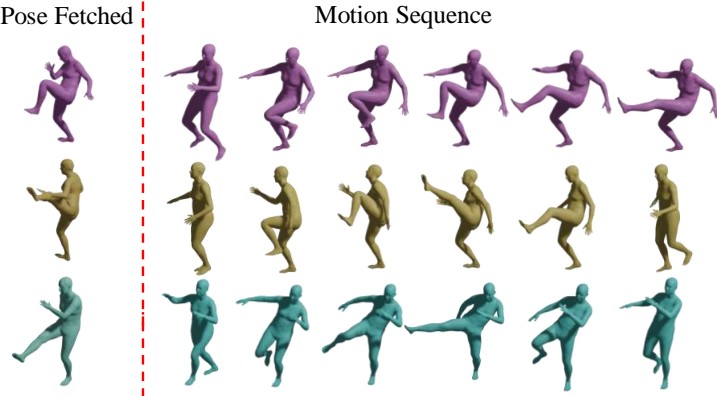

Figure 3: Fetching different poses to guide the motion generation from the same text *A person kicks*.

We demonstrate that our pose guidance effectively influences the motion generated in Figure 3 with the results of generating motion from the same text input but guided by different poses fetched. The text prompt is set as *A person kicks*, while poses used as guidance are chosen differently. We can see that each motion generated is apparently related to the pose given, and does not appear to be an average version affected by different poses the model may have encountered in the training process.

### 5 CONCLUSION

We introduce Pose-Guided Text to Motion (PG-T2M), which constructs a pose memory that stores mappings between text clusters of the semantically similar single actions and their pose information. We split the text into sub-sentences to handle the entangled verbs in complicated sentences and help build a concrete mapping between single action descriptions and the motions. We propose to automatically generate pose features for each sub-sentence to guide the generation of motions. Quantitative experiments affirm the superior performance of our method compared to existing techniques in text-driven motion generation.

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

# A    APPENDIX

In this supplementary material, we present necessary additional information and take a deeper dive into our approach. In Appendix A.1, we present additional experiments. We make more discussions on its performance with single-action texts, repeat the ablation studies on MoMask, and compare the performance of different pose extractors. In Appendix A.2, we show more visual results and discuss the zero-shot ability of pose-guided motion generation. In Appendix A.3, we provide more implementation details on how we apply our design principles on different baselines, along with the introduction of each evaluation metric we use.

## A.1    ADDITIONAL EXPERIMENTS AND DISCUSSIONS

### A.1.1    EXPERIMENTS ON RESPLIT DATASET

Here we compare the performance of our model, MDM (Tevet et al., 2023) and MotionDiffuse (Zhang et al., 2022) across varied test set splits. The aim is to investigate how the performance is impacted by the varying numbers of actions present in the text input.

Texts in training data may include multiple actions, making it difficult for the model to learn the concrete relationship between each action description and motion, resulting in bad performance

Table 6: Result on KIT (Plappert et al., 2016) dataset, texts including one single action.

| Method | Single action in KIT | | |
| --- | --- | --- | --- |
| | R% (top 3)↑ | FID↓ | MM Dist↓ |
| Ground truth | 77.0 | 0.051 | 2.618 |
| MDM | 37.7 | 0.506 | 9.592 |
| Ours with MDM | **42.1** | **0.361** | **9.497** |
| MotionDiffuse | 73.2 | 1.915 | 2.861 |
| Ours with MotionDiffuse | **76.2** | **0.109** | **2.703** |

Table 7: Result on KIT (Plappert et al., 2016) dataset, texts including more than one action.

| Method | Multiple actions in KIT | | |
| --- | --- | --- | --- |
| | R% (top 3)↑ | FID↓ | MM Dist↓ |
| Ground truth | 75.7 | 0.078 | 3.042 |
| MDM | **49.6** | 0.893 | 8.406 |
| Ours with MDM | 49.0 | **0.344** | **8.398** |
| MotionDiffuse | 74.3 | 2.076 | **3.068** |
| Ours with MotionDiffuse | **75.2** | **0.122** | 3.084 |

Table 8: Result on HumanML3D (Guo et al., 2022) dataset, texts including one single action.

| Method | Single action in HumanML3D | | |
| --- | --- | --- | --- |
| | R% (top 3)↑ | FID↓ | MM Dist↓ |
| Ground truth | 76.5 | 0.003 | 3.034 |
| MDM | 57.0 | **0.688** | 6.355 |
| Ours with MDM | **60.9** | 0.690 | **6.203** |
| MotionDiffuse | 76.2 | 0.627 | 3.026 |
| Ours with MotionDiffuse | **79.4** | **0.154** | **2.994** |

Table 9: Result on HumanML3D (Guo et al., 2022) dataset, texts including more than one action.

| Method | Multiple actions in HumanML3D | | |
| --- | --- | --- | --- |
| | R (top 3)↑ | FID↓ | MM Dist↓ |
| Ground truth | 78.7 | 0.002 | 3.032 |
| MDM | 61.5 | **0.484** | 5.336 |
| Ours with MDM | **65.8** | 0.647 | **5.126** |
| MotionDiffuse | 76.0 | 0.638 | **3.185** |
| Ours with MotionDiffuse | **78.0** | **0.166** | 3.203 |

on single-action data. We conduct experiments on texts including different numbers of actions on KIT (Plappert et al., 2016) and HumanML3D (Guo et al., 2022) dataset using MDM (Tevet et al., 2023) and MotionDiffuse (Zhang et al., 2022). Here we present a complete comparison in Tables 6 to 9. 65.45% of the data in KIT is single-action and 36.97% of the data in HumanML3D is single-action. As we can find in the tables: 1) MDM performs worse on single action on both KIT and HumanML3D, illustrating that MDM does not understand the concrete relationship between each single action description and the motion well. 2) Our method on MDM wins a great gain on single-action data, improving R-top3 by 4.4%, FID by 0.145 MM Dist by 0.095 on KIT, and R-top3 by 3.9%, MM Dist by 0.152 on HumanML3D. Meantime, the ability to multiple-action data is also enhanced. This proves that our method by splitting text into single actions and involving action pose features can enhance the model's understanding of the motion corresponding to a single action. 3) MotionDiffuse has a more balanced performance on both datasets, yet our method still improves its performance on each dataset, which shows the strong universality and portability of our method.

### A.1.2 ABLATION STUDIES WITH MOMASK AS THE BASELINE

Table 10: Ablation study on choices of pose memory usage on HumanML3D based on MoMask.

| Temporal encoder | Random selection | Text splitting | R% (top 1)↑ | R% (top 2)↑ | R% (top 3)↑ | FID↓ | MM Dist↓ |
|---|---|---|---|---|---|---|---|
| ✓ | ✓ | ✓ | **53.1** | **72.0** | **81.5** | 0.064 | **2.908** |
| ✗ | ✓ | ✓ | 52.4 | 71.5 | 80.8 | 0.061 | 2.953 |
| ✓ | ✗ | ✓ | 52.5 | 71.5 | 81.1 | **0.059** | 2.927 |
| ✓ | ✓ | ✗ | 52.7 | 71.8 | 81.3 | 0.067 | 2.923 |

We repeat the ablation studies based on MoMask (Guo et al., 2023). As shown in Table 10, the results prove the effectiveness of our designs for pose memory usage. In the main text, we choose MDM as the baseline in ablation studies mainly because it is a simple baseline without heuristic designs like a hierarchical quantization scheme or residual transformer, which can better reflect the effectiveness of each module we propose. On the other hand, the ablations based on MoMask also prove the generalization and effectiveness of our method.

### A.1.3 ABLATION STUDIES ON THE POSE EXTRACTOR

Table 11: Ablation study on choices of pose extractors.

| Pose Extractor | R% (top 3)↑ | FID↓ | MM Dist↓ |
|---|---|---|---|
| PyMAF-X | 64.5 | 0.689 | 5.355 |
| DecoMR | 63.1 | 0.730 | 5.367 |
| METRO | 64.2 | **0.661** | 5.358 |
| OSX-UBody | **64.7** | 0.663 | **5.338** |

We have compared the performance of models using different pose extractors in Table 11. Although there are slight performance differences when using different pose extractors, these variations are not substantial. Additionally, existing available pose extractors have already demonstrated good performance in pose reconstruction.

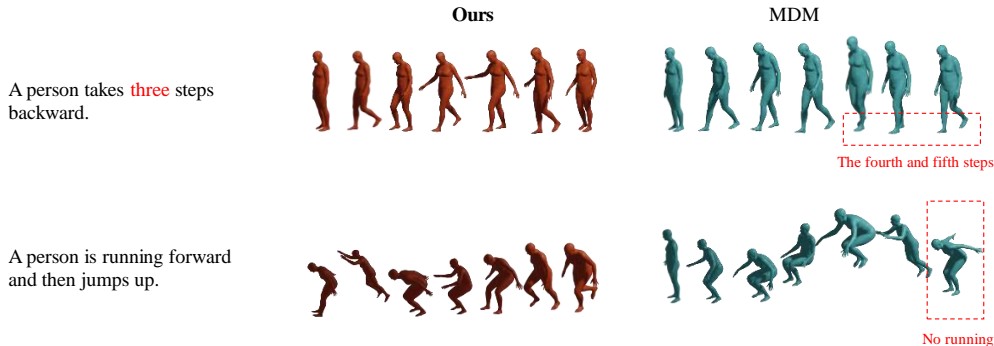

Figure 4: The comparison between original MDM (Tevet et al., 2023) and our MDM-based model. We choose two examples to show how our model performs better in fine-grained or coupled actions. In the first example, motion frames are placed from left to right; in the second one, motion frames are placed from right to left.

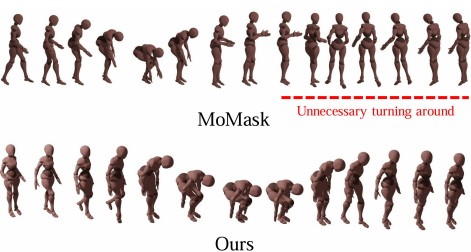

Figure 5: The text prompt is: *The person walks forward, then stops and bends down to pick up something.*

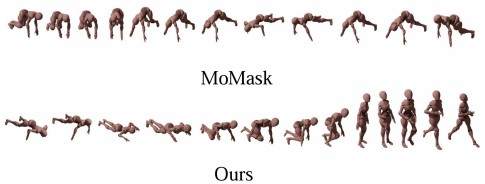

Figure 6: The text prompt is: *The person is doing push-ups, then stands up and runs forward.*

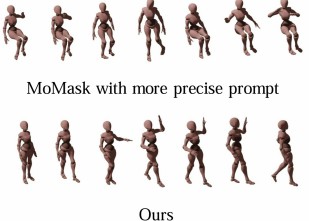

Figure 7: The original text prompt is: *A person walks while raising his hand up.* The enhanced text prompt is: *A person walks while raising his hand up; during the process, the person moves to the south, his left forearm moves to the body's left up.* The enhancement comes from SemanticBoost.

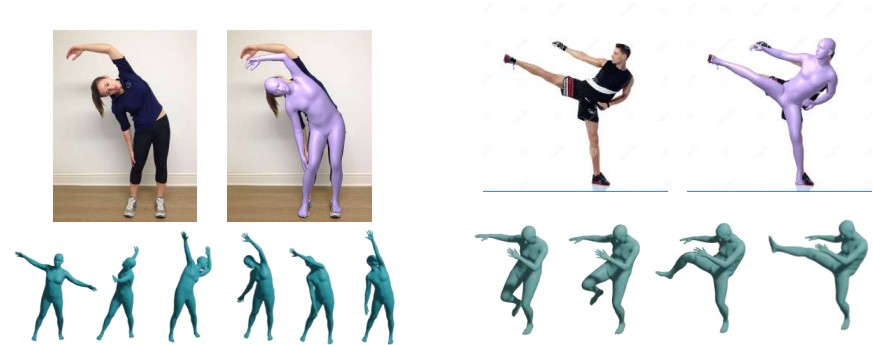

(a) The text is *An individual is doing stretching.*    (b) The text is *A person kicks.*

Figure 8: Using an unseen pose extracted from an image out of our pose memory to guide the generation. In each subfigure, **upleft:** the novel image from the Internet; **upright:** the unseen pose extracted from the image; **down:** motion generated guided by the unseen pose.

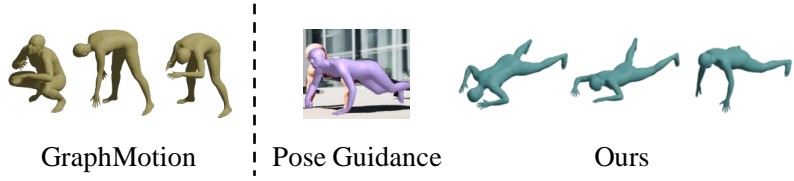

GraphMotion     Pose Guidance     Ours

Figure 9: Unseen text condition: *A human is doing push-ups.*

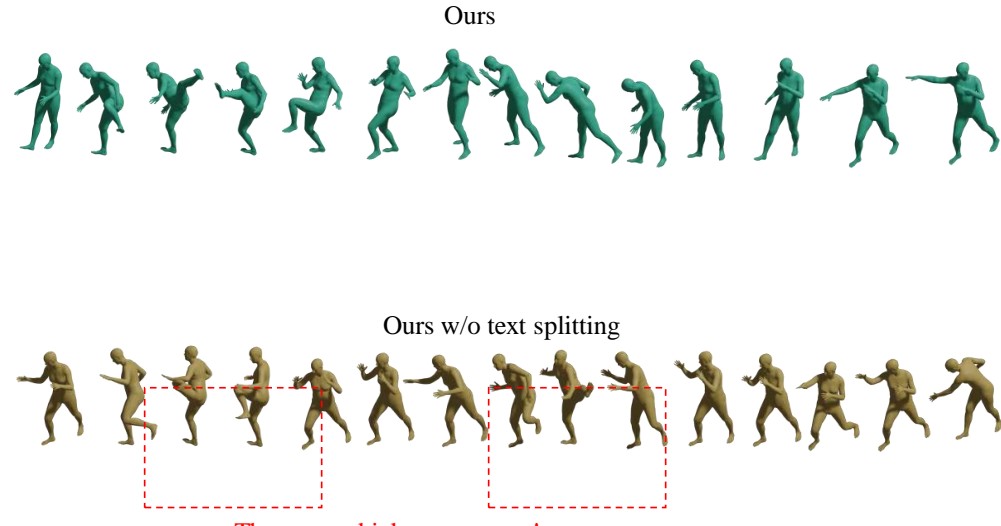

Figure 10: The comparison between our model based on MDM and the one without text splitting module. The text input is *A person kicks with his left leg and then punches to the front*. Motion frames are placed from left to right.

## A.2 ADDITIONAL VISUALIZATIONS

### A.2.1 QUALITATIVE COMPARISON

We compare our results with MDM (Tevet et al., 2023) and present them in Figure 4. We explored the performance of our model on texts including multiple actions and fine-grained action descriptions.

Compared with MDM, our model performs better when encountering multiple-action-included texts. And for those few-action texts, our model generates more concrete motion sequences. We can observe that: 1)MDM cannot understand specific figures like *three*, while ours control the motion generated better in a fine-grained way. As in our first example in Figure 4, MDM just keeps *stepping backward*, ignoring how many steps have been taken. Our result is much more accurate, the avatar takes exactly *three* steps back. 2)MDM is easily distracted by a certain verb while there is more than one included in a sentence, while ours separates them correctly. As is shown in the second example in Figure 4, MDM neglects the action *running* and only generates a *jumping* motion. Our model avoids this mistake. In summary, our pipeline helps generate more reasonable, fine-grained, and concrete motions.

We add three more qualitative visualizations comparing MoMask (Guo et al., 2023) and our results. Figure 5 demonstrates that MoMask cannot correctly handle it when multiple actions are involved in textual prompts, where MoMask incorrectly generates a motion clip of turning around. Figure 6 presents that when MoMask meets a novel verb-*push-up*-it will be confused and unable to generate the subsequent motion of standing up and running forward. As shown in Figure 7, we manually pro-

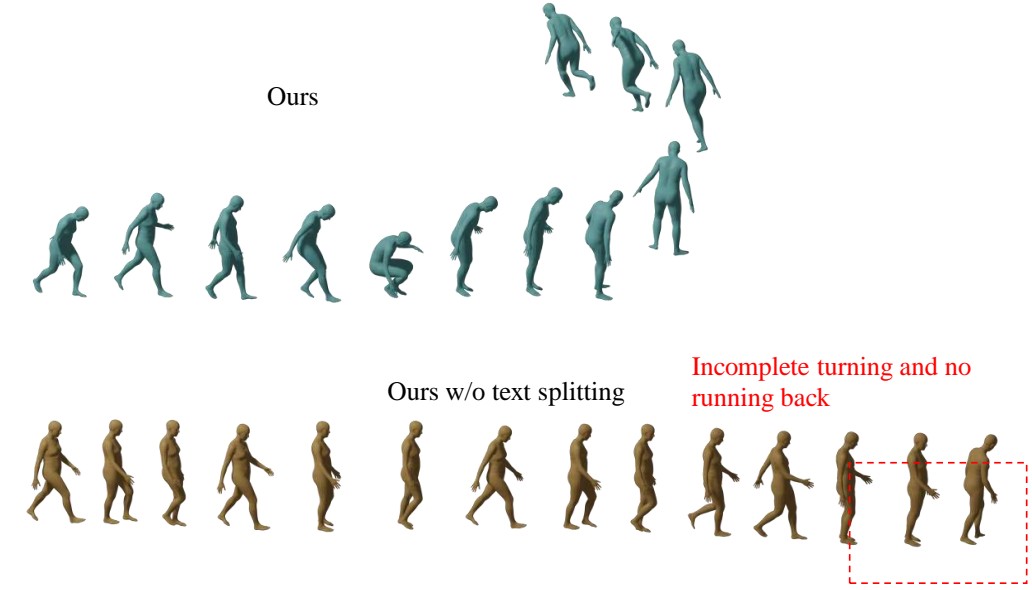

Figure 11: The comparison between our model based on MDM and the one without text splitting module. The text input is *A person walks forward and suddenly squats down, then he turns around and runs back*. Motion frames are placed from left to right.

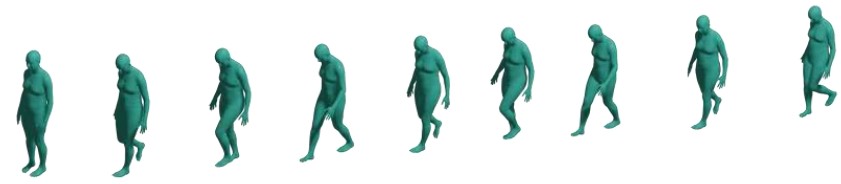

Figure 12: More visualized samples. The text input is *A person takes six steps backward*. Motion frames are placed from left to right.

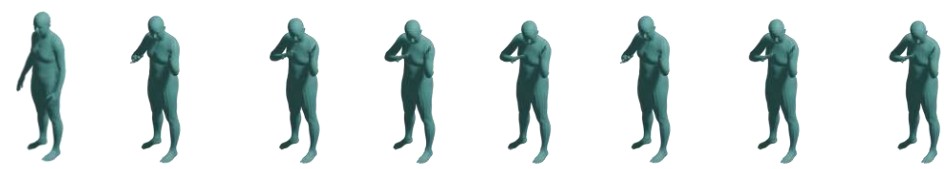

Figure 13: More visualized samples. The text input is *A person is playing the violin*. Motion frames are placed from left to right.

vide the baseline with a more precise prompt and compare the generated motions with our method. The more precise prompt comes from SemanticBoost. We can observe that the MoMask performs badly with the precise prompt and seems confused by the complex sentence. The information pro-

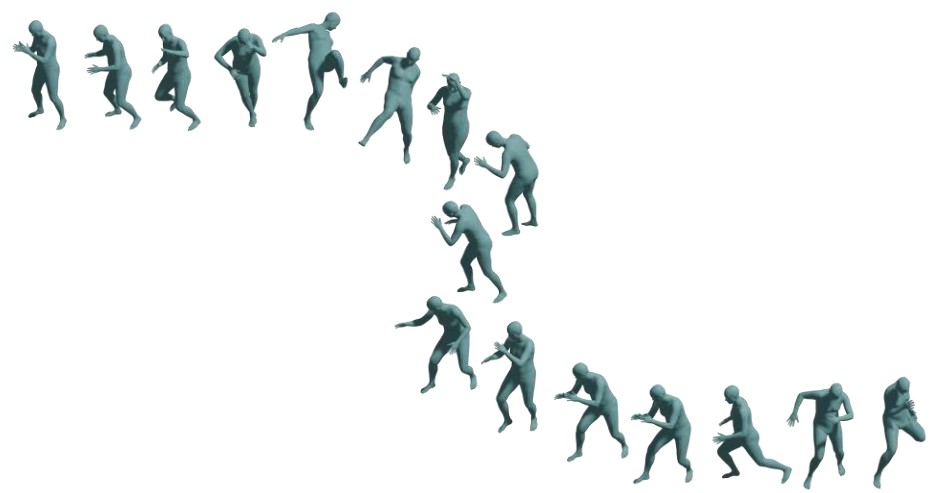

Figure 14: More visualized samples. The text input is *A person is playing soccer.* Motion frames are placed from left to right.

vided by a more precise prompt is still hard to extract. Better prompts alone cannot handle the problem of action entanglement or help the model identify the concrete one-to-one association.

### A.2.2 VISUALIZATION OF ZERO-SHOT POSE GUIDANCE

We have demonstrated in the main text that the poses in our pose memory can control the motion generated. Further, we also find that our model demonstrates a certain zero-shot generalization ability in using unseen poses extracted from novel images to guide the motion generation during inference as we show in Figure 8. We select some images from the Internet and extract poses from them which is totally new to the model. Using these unseen poses as guidance, the model can still generate motions including these poses.

Furthermore, we use a textual prompt *A human is doing push-ups* not existing in the train set along with an unseen figure to successfully generate a motion. The result shown in fig. 9 proves that our method supports to control motion by user-specified pose images and can generalize to unseen poses. This indicates that users can specify poses to more precisely control the generated motions.

### A.2.3 VISUALIZATIONS OF ABLATION STUDIES

In Figures 10 and 11, we verify the effectiveness of the text splitting module. Specifically, we can see from Figure 10 that without the text splitting module, the motion generated gets confused with the temporal connections of each action and incorrectly repeats the motion once more. According to Figure 11, the exclusion of the text splitting module causes the model to generate incomplete motions or neglect certain actions.

In Figures 12 to 14, we present more visualizations of our models. In Figure 12, we find that our model has an accurate sense of details, like the number of steps it needs to take. In Figure 13, we present another example to show the ability of our model to generate various motions like playing the instruments. We use Figure 14 to prove that the utilization of pose-feature-from-text does not ruin the ability to generate an abstract and complex motion like playing soccer.

### A.3 EXPERIMENT DETAILS

In this section, we first illustrate how we apply our pose-guided conditioning on MotionDiffuse (Zhang et al., 2022) in Appendix A.3.1. Later, we show the computation resources we consume in Appendix A.3.2. We then describe the detailed evaluation process in Appendix A.3.3.

### A.3.1 ARCHITECTURE DETAILS

In the main text, we outlined the implementation details of applying our design principles to the baseline MDM (Tevet et al., 2023). In this section, we elaborate on the application of our pipeline to MotionDiffuse (Zhang et al., 2022). Unlike MDM (Tevet et al., 2023), which utilizes only sentence-level text features supplemented by timestep embeddings as singular guidance, MotionDiffuse incorporates token-level text features for cross attention with motion frame representations. To seamlessly incorporate our pose memory design into MotionDiffuse with minimal alterations, we begin by constructing the pose memory (involving the splitting of raw texts and encoding of pose features) as proposed in the main text. Subsequently, for the encoded pose feature, we incorporate it into each token-level text feature of MotionDiffuse as conditions to guide the motion generation. Notably, the diffusion process of MotionDiffuse remains unaltered throughout this integration process.

### A.3.2 COMPUTATION RESOURCES

We use only one single *NVIDIA GeForce RTX 4090* GPU when training the MDM-based model with a batch size of 64. We use 2 *Tesla A100* GPUs with 256 samples on each GPU when training the MotionDiffuse-based model. We use one single *NVIDIA GeForce RTX 4090* GPU when training the MoMask-based model with a batch size of 512 for rvq training, a batch size of 64 for transformer training.

### A.3.3 EVALUATION METRIC DETAILS

We follow the standard test protocol as adopted by (Guo et al., 2022) to evaluate all methods with five different metrics. In this section, we provide more details on how these metrics are calculated. Features are first extracted from both the generated motions and ground truth motions by the pre-trained motion encoder, denoted as $f_{gen}, f_{gt}$. The text features are denoted as $f_t$

**Frechet Inception Distance(FID).** FID is widely used in generation tasks to evaluate the overall quality. Specifically, FID is calculated between ground truth and generated distributions to measure the similarity. A lower FID is better, which means the overall generated motions are more similar to the ground truth. We use

$$
\begin{aligned}
FID = &\|\mu_{gt} - \mu_{gen}\|_2^2 + \\
&Trace(\Sigma_{gt} + \Sigma_{gen} - 2(\Sigma_{gt}\Sigma_{gen})^{1/2})
\end{aligned}
\tag{7}
$$

to compute the FID, where $\mu_{gt}, \mu_{gen}$ represents mean of $f_{gt}, f_{gen}$ and $\Sigma$ is the covariance matrix.

**Multimodal Distance(MM Dist).** Multimodal distance is used to measure the difference between the text feature and the motion feature. A lower MM Dist is better, which represents that the motions generated are closer to the texts given. We use

$$
Dist = \frac{1}{N} \sum_{i=1}^{N} \|f_t - f_{gen}\|_2
\tag{8}
$$

where $N$ is the length of the motion.

**R-precision.** For each text input, we choose the 31 other test text inputs in the same batch during evaluation and then the multimodal distance will be computed between the generated motion and these 32 texts. R-precision represents the average accuracy of matching (the matched text-motion pairs have the smallest multimodal distance). A high R-precision is better, which means that the model can generate motions close to its corresponding text description and easily be distinguished from others.

**Diversity.** We use diversity to measure the variance of the whole motion sequences across the dataset. A higher diversity is better, meaning the motion generated is more varied. $S$ pairs of motions(we refer to their features using $f_{i,1}, f_{i,2}$) are randomly sampled from the generated motions, and we calculate diversity using

$$
D = \frac{1}{S} \sum_{i=1}^{S} \|f_{i,1} - f_{i,2}\|_2
\tag{9}
$$

Here we set $S = 300$ following HumanML3D (Guo et al., 2022).

