# OpenReview forum: "Pose-guided Motion Diffusion Model for Text-to-motion Generation"
_ICLR.cc/2025/Conference — ICLR 2025 Conference Withdrawn Submission_

### Official Review · Reviewer_VNNS · 2024-11-01

**Soundness:** 2
**Presentation:** 2
**Contribution:** 2
**Rating:** 3
**Confidence:** 5

**Summary:**

This paper proposes to divide the sentences into sub-sentences containing one single verb and make the model learn the specific mapping from one single action description to its motion. The author also utilizes the off-the-help text-to-motion model to get more data. The author finally get SOTA results on Humanml3d and KIT dataset.

**Strengths:**

1. The author utilizes the off-the-shelf T2I model to enrich the dataset.
2. The model get SOTA results.

**Weaknesses:**

1. The authors try to solve the complex text problem in the motion generation dataset. However, the text and other modality ambiguous problems also exist in other domains, like videos or images. They solve the problem using a better text encoder and large various texts. The method of this paper is not universal and hard to scale.
2. The text of text-motion dataset is actually not as complicated as the video dataset. I think the motivation is not that solid.

Overall, from my perspective, this paper is not suitable for ICLR.

**Questions:**

1. Did the author try to use a better text encoder, like T5-xl or T5-xxl? Lots of paper proves that the clip text encoder is not good enough for extracting text embedding.
2. Does the generated motion follow the given pose? The way you are using poses can not utilize the information well. The given text can only see some relevant texts and poses, which lost a lot of information. Why not pretrain a model with both poses and motions using some mechanism like causal properties?

---

### Official Review · Reviewer_YGkN · 2024-11-01

**Soundness:** 2
**Presentation:** 2
**Contribution:** 1
**Rating:** 5
**Confidence:** 5

**Summary:**

The authors present a pose-guided motion diffusion model (PG-T2M) that incorporates pose priors extracted from images generated by text-to-image diffusion models (i.e., Stable Diffusion 2.1). A pose memory is constructed to retrieve pose information for each action, which is then combined with text inputs to enhance model performance.

**Strengths:**

The authors skillfully build a pose information memory by leveraging a text-to-image generation process followed by pose extraction from the generated images, supplying enhanced prior knowledge for motion generation. Quantitative results demonstrate that the proposed method benefits multiple base models, such as MoMask and MDM.

**Weaknesses:**

- The easiest way to boost the model’s text alignment would be to improve the text encoder. If you think CLIP’s sentence-level feature is weak, why not try using token-level features? And if token-level features don’t perform well either, could you show some comparison experiments to back it up? **Do not complicate a simple problem, it’ll just lose the practical value needed for acceptance.**
- In your demo, I only see some low-quality text2motion visualizations, so where is the most important part for **pose-guided** text2motion?

**Questions:**

Missing cites:
- MotionGPT: Human Motion as a Foreign Language
- MotionLCM: Real-time Controllable Motion Generation via Latent Consistency Model
- StableMoFusion: Towards Robust and Efficient Diffusion-based Motion Generation Framework
- Mofusion: A framework for denoising-diffusion-based motion synthesis

---

### Official Review · Reviewer_AbJZ · 2024-11-02

**Soundness:** 3
**Presentation:** 3
**Contribution:** 2
**Rating:** 5
**Confidence:** 5

**Summary:**

This paper introduces a pose memory designed better to align sub-action text with poses in human motion generation. The proposed pose memory is used for retrieving text embeddings for conditioning human motion generation, ensuring better alignment between text and pose. Experimental results demonstrate that the pose memory can be applied broadly across diffusion-based methods for text conditioning, improving performance and achieving state-of-the-art results in human motion generation on the HumanML3D and KIT datasets.

**Strengths:**

* The motivation for this work is clearly articulated in the introduction.
* Experiments show that the proposed pose memory generalizes well, providing a performance boost across several diffusion-based motion generation methods.
* The proposed method achieves state-of-the-art performance in human motion generation on the HumanML3D and KIT datasets, which is impressive.

**Weaknesses:**

* Constructing a memory bank is highly dependent on the performance of other models, such as those for image synthesis and pose extractor, and pose encoder, which may limit the reproducibility of this work.
* The technical contribution, while valuable, is limited in novelty. Constructing a memory bank is closer to data augmentation using pretrained networks from other domains than to a novel techincal advancement.
* Is clustering necessary to build the pose memory? Performance seems to increase with more clusters—what is the baseline performance without clustering?

**Questions:**

* Would it be helpful to compare hybrid features, text features, and pose features for conditioning to justify the design of pose guided conditioning?
* Given the complex steps and dependency on external libraries to build the pose memory, will the code and memory bank for each dataset be published to aid reproducibility?
* In line 249, what are “proto-agent” and “proto-patient”? For clarity, please briefly explain these terms in the paper.
* Are the results in Table 3 the same as those in Figure 2(b)?
* Is the temporal encoder trained jointly with the motion diffusion network?

---

### Note · Authors · 2024-11-15

I have read and agree with the venue's withdrawal policy on behalf of myself and my co-authors.